# Creatinine-to-Cystatin C Ratio Combined with FIB-4 and ELF for Noninvasive Fibrosis Assessment in MASLD

**DOI:** 10.3390/ijms26199560

**Published:** 2025-09-30

**Authors:** Masafumi Oyama, Tadashi Namisaki, Akihiko Shibamoto, Satoshi Iwai, Masayoshi Takami, Yuki Tsuji, Yukihisa Fujinaga, Hiroaki Takaya, Takashi Inoue, Norihisa Nishimura, Shinya Sato, Koh Kitagawa, Kosuke Kaji, Akira Mitoro, Kiyoshi Asada, Hiroyuki Masuda, Junichi Hanatani, Hitoshi Yoshiji

**Affiliations:** 1Department of Gastroenterology, Nara Medical University, 840 Shijo-cho, Kashihara 634-8521, Nara, Japanmitoroak@naramed-u.ac.jp (A.M.);; 2Department of Evidence-Based Medicine, Nara Medical University, 840 Shijo-cho, Kashihara 634-8522, Nara, Japan; 3Clinical Research Center, Nara Medical University, 840 Shijo-cho, Kashihara 634-8522, Nara, Japan; kasada@naramed-u.ac.jp

**Keywords:** MASLD, the creatinine-to-cystatin C ratio, non-invasive tool, liver fibrosis

## Abstract

The creatinine-to-cystatin C ratio (CCR), a surrogate for skeletal muscle mass, may also be associated with liver fibrosis due to the strong link between sarcopenia and liver disease progression. We aimed to evaluate the utility of CCR as a noninvasive marker of liver fibrosis in metabolic-dysfunction-associated steatotic liver disease (MASLD). This retrospective study included 104 patients with biopsy-proven MASLD. CCR was calculated using serum creatinine and cystatin C levels. Liver fibrosis was staged histologically (F0–F4), and skeletal muscle mass was assessed using the skeletal muscle index (SMI) on computed tomography. Associations between CCR and liver fibrosis, SMI, and nonalcoholic fatty liver disease activity score were analyzed. ROC analysis evaluated CCR performance alone and in combination with FIB-4 and enhanced liver fibrosis (ELF) scores. CCR values were significantly lower in patients with significant fibrosis (≥F2). The AUROC of CCR for detecting ≥F2 fibrosis was 0.621 (95% CI: 0.509–0.733), with an optimal cutoff of 0.664. CCR alone yielded an AUC of 0.815 for predicting ≥F2 fibrosis. Combining CCR with FIB-4 and ELF substantially improved diagnostic accuracy, increasing the AUROC from 0.621 (CCR alone) to 0.820 for the combined model. CCR correlated positively with SMI (r = 0.451, *p* < 0.001). CCR is a simple, cost-effective biomarker reflecting muscle mass and liver fibrosis in MASLD. Combining CCR with established markers may enhance risk stratification and reduce the need for liver biopsy in selected cases.

## 1. Introduction

Metabolic-dysfunction-associated steatotic liver disease (MASLD), a recently defined entity, has replaced nonalcoholic fatty liver disease (NAFLD) to better reflect the close relationship between hepatic steatosis and metabolic risk factors [1]. MASLD is diagnosed when hepatic steatosis is present along with at least one cardiometabolic risk factor, including obesity, type 2 diabetes mellitus, dyslipidemia, or hypertension [2]. This redefinition emphasizes the metabolic basis of steatosis and its potential to progress to advanced liver disease [3]. The global burden of MASLD is substantial and rising, with prevalence estimates reaching 25–30% in the general population and higher in high-risk groups [4]. MASLD encompasses a spectrum from simple steatosis to steatohepatitis and cirrhosis [4]. Among its histological features, fibrosis stage is the strongest predictor of liver-related and overall mortality. Patients with stage ≥F2 fibrosis have a significantly increased risk of developing hepatocellular carcinoma, liver failure, and extrahepatic complications, such as cardiovascular disease [5].

Traditionally, liver biopsy has been the reference standard for fibrosis assessment. However, its invasiveness, cost, sampling variability, and associated discomfort limit its suitability for routine use [6]. Consequently, noninvasive tools, such as the FIB-4 index and the enhanced liver fibrosis (ELF) score, have gained popularity. The FIB-4 index is based on age, AST, ALT, and platelet count [7]. The ELF score incorporates direct markers of extracellular matrix turnover, including hyaluronic acid, procollagen III N-terminal peptide (PIIINP), and tissue inhibitor of metalloproteinase-1 (TIMP-1). Both are useful for screening and stratifying fibrosis risk, but they have limitations, particularly in patients with indeterminate results.

Evidence increasingly highlights the role of sarcopenia—the progressive loss of skeletal muscle mass and strength—in chronic liver disease. Sarcopenia is independently associated with liver fibrosis, hepatic decompensation, and poor clinical outcomes [8]. In MASLD, metabolic dysfunction may promote both hepatic fibrogenesis and muscle wasting through shared mechanisms, including insulin resistance, systemic inflammation, and oxidative stress [9]. Moreover, muscle-derived myokines, including irisin and myostatin, influence hepatic metabolism and fibrosis pathways [10]. Markers reflecting muscle status and liver pathology may therefore offer unique insights into disease severity and prognosis [11].

The creatinine-to-cystatin C ratio (CCR) is a laboratory-based marker originally proposed as a surrogate for skeletal muscle mass. This serum ratio has emerged as a practical and accessible indicator of muscle mass (a proxy for sarcopenia) and has demonstrated prognostic value for mortality and other adverse outcomes in both hospitalized [12] and critically ill patients [13]. Nevertheless, CCR should be regarded as a complementary tool rather than a replacement for standard diagnostic approaches to sarcopenia, such as computed tomography (CT) imaging or grip strength assessment, particularly in patients with stable renal function [14]. Its primary utility lies in screening, where it may help identify high-risk individuals who could benefit from further evaluation or timely intervention [15].

Creatinine, a breakdown product of creatine phosphate in muscle tissues, and cystatin C, produced by all nucleated cells and filtered by the kidney independently of muscle mass, together form the basis for CCR. This ratio has emerged as a practical tool for muscle mass assessment, particularly in older adults [12]. Studies have demonstrated its value in detecting sarcopenia, predicting hospitalization, and assessing mortality risk [16,17,18,19]. More recently, low CCR has been linked to MASLD severity and advanced fibrosis [13]. CCR reflects skeletal muscle mass and handgrip strength, and its association with the severity of MASLD-related fibrosis appears to be indirectly mediated by progressive muscle loss. However, to date, no study has systematically examined the relationship between CCR and histologically confirmed liver fibrosis in MASLD.

This study aimed to fill this gap by evaluating CCR as a noninvasive marker for predicting liver fibrosis in MASLD. This study also evaluated its association with muscle mass and the NAFLD activity score (NAS), including steatosis, lobular inflammation, and ballooning. Finally, we assessed whether combining CCR with established fibrosis scores could improve diagnostic performance for detecting significant fibrosis.

## 2. Results

### 2.1. Patient Characteristics

A total of 104 patients with biopsy-confirmed MASLD were included in the final analysis (Table 1). The median age was 62.9 years (IQR: 50.0–67.8), and 73.1% were male. The median BMI was 27.3 kg/m^2^, with nearly 80% of the cohort meeting the Japanese criteria for obesity (BMI ≥ 25 kg/m^2^). Metabolic comorbidities were highly prevalent, with type 2 diabetes mellitus present in 57.7% (60/104), hypertension in 53.8% (56/104), and dyslipidemia in 38.5% (40/104) of patients. The median HbA1c was 6.3% (IQR: 5.7–7.0%), and 57.7% of the participants were receiving therapy with either a dipeptidyl peptidase-4 inhibitor, a sodium–glucose cotransporter-2 inhibitor, or a glucagon-like peptide-1 receptor agonist. Liver function tests revealed a mean AST of 48 IU/L (IQR: 34–73) and ALT of 64 IU/L (40–95), while the median platelet count was 19.0 × 10^4^/μL. The median FIB-4 index was 1.79 (IQR: 1.08–3.19), and the mean ELF score was 10.06 (IQR: 9.22–11.0). The median serum creatinine level was 0.69 mg/dL (IQR: 0.59–0.84) and the median cystatin C level was 1.03 mg/dL (IQR: 0.89–1.16), yielding a median CCR of 0.658 (IQR: 0.581–0.816).

### 2.2. Histology

Histological analysis showed that 13.5% of the patients had no fibrosis (F0), 23.1% had stage F1, 26.9% had stage F2, 25.0% had stage F3, and 11.5% had cirrhosis (F4), resulting in 63.5% of the cohort having advanced fibrosis (F ≥ 2). The mean NAS was 4.7 ± 1.2, and 62.5% of the patients met the histological criteria for steatohepatitis (Figure 1). Patients with significant fibrosis exhibited significantly lower CCR values than those without significant fibrosis (0.736 vs. 0.634, *p* < 0.05). CCR correlated significantly negatively with fibrosis stage (Spearman’s rho = −0.24, *p* < 0.05) (Figure 1). CCR was significantly higher in patients with ≤F2 than in those with >F2 (Figure 2). ROC curve analysis for CCR in detecting significant fibrosis yielded an area under the receiver operating characteristic curve (AUROC) of 0.621 (95% confidence interval [CI]: 0.509–0.733), with an optimal cutoff of 0.664 determined by the Youden index, corresponding to a sensitivity of 65.1% and a specificity of 66.3% (Figure 2). Metabolic dysfunction-associated steatohepatitis (MASH) was diagnosed in 69.2% (72/104) of the patients. Inter-observer variability in histology scoring was carefully reviewed through collaborative discussions between hepatologists and pathologists.

### 2.3. Diagnostic Performance of FIB-4 and ELF Scores and Their Combination with CCR

Although the discriminative performance of CCR alone was modest, its combination with established fibrosis markers enhanced the predictive accuracy. The AUROC was 0.798 for FIB-4 and 0.815 for ELF (Figure 3 and Figure 4); when CCR was added to these scores, the AUROC increased to 0.820 (Figure 5). Combining CCR with FIB-4 and ELF substantially improved diagnostic accuracy, increasing the AUROC from 0.621 (CCR alone) to 0.820 for the combined model.

### 2.4. Decision Tree Analysis

Decision tree analysis revealed that patients with an FIB-4 index < 1.3 were predominantly classified as having mild fibrosis (F < 2; 50/66 cases). In contrast, patients with an FIB-4 index ≥ 1.3 were further stratified using the ELF score. An ELF score ≥ 9.87 identified a high proportion of significant fibrosis (F ≥ 2; 47/56 cases). For patients with ELF ≥ 9.87, a CCR < 0.664 indicated significant fibrosis in 31/34 cases. Conversely, among patients with ELF < 9.87, a CCR ≥ 0.664 indicated no significant fibrosis, with mild fibrosis confirmed in 7/10 cases (Figure 6). Overall, in patients with positive FIB-4 results (≥1.3), the addition of ELF and CCR measurements increased the positive predictive value from 76% to 91%.

The FIB-4 index remains a practical screening tool for identifying advanced fibrosis (F ≥ 2) in nonalcoholic steatohepatitis (NASH) within large populations. However, despite its relatively high sensitivity, specificity is often low, particularly with the “gray zone” range of 1.3–2.67, where nonfibrotic individuals may present with elevated FIB-4 values (≥1.3). This limitation can lead to false-positive classification. Therefore, incorporating ELF testing and CCR measurement in individuals with positive FIB-4 results could reduce misclassification and improve diagnostic precision.

In patients with eGFR ≥ 60, the decision tree analysis showed that those with an FIB-4 index < 1.3 were predominantly classified as having mild fibrosis (F < 2; 27/37 cases). By contrast, patients with an FIB-4 index ≥ 1.3 were further stratified using the ELF score. An ELF score of ≥9.87 identified a high proportion of significant fibrosis (F ≥ 2; 38/45 cases). Among them, a CCR of <0.664 indicated significant fibrosis in 31/33 cases. Conversely, for patients with ELF < 9.87, a CCR of ≥0.664 was associated with the absence of significant fibrosis, with mild fibrosis confirmed in 7/10 cases (Figure 7). Overall, in patients with positive FIB-4 results (≥1.3), incorporating ELF and CCR measurements increased the positive predictive value from 75% to 94%. Notably, this approach proved particularly useful for identifying patients with fibrosis stage ≤F2 among those with eGFR ≥ 60 but was less effective in patients with 30 ≤ eGFR < 60 (Figure 8).

### 2.5. Relationship Between Skeletal Muscle Mass, Steatosis/Inflammation, and CCR

Skeletal muscle mass was evaluated in 92 patients, and CCR showed a significant positive correlation with SMI (r = 0.451, *p* < 0.001) (Figure 9). In contrast, CCR was not significantly associated with steatosis grade, lobular inflammation score, or hepatocellular ballooning (Appendix A). Although there was a trend toward lower CCR values in patients with ballooning, the difference was not statistically significant (*p* = 0.0533). These findings suggest that CCR is more closely linked to chronic structural changes, such as fibrosis, rather than to the active inflammatory features of MASLD.

### 2.6. Relationship Between NAS and Cystatin C

Serum Cystatin C levels, when considered alone, were not significantly associated with the severity of fibrosis, steatosis, inflammation, or hepatocellular ballooning (Appendix A).

## 3. Discussion

This study evaluated the potential of serum CCR as a noninvasive biomarker for liver fibrosis in patients with MASLD and found that lower CCR values correlated significantly with both advanced liver fibrosis and reduced skeletal muscle mass. This supports its potential role as a dual-purpose indicator of hepatic and muscular health. This study is among the first to investigate this relationship in a well-characterized MASLD cohort with both paired liver biopsy and CT-based skeletal muscle evaluation. Although the discriminative performance of CCR alone was modest compared with conventional noninvasive tests, such as FIB-4 and ELF scores, combining CCR with these established indices improved the accuracy of identifying patients with histologically confirmed fibrosis stage ≥F2 [14]. This suggests that CCR offers added value to cases where traditional scores yield intermediate or indeterminate results. The strong positive correlation between CCR and skeletal muscle index (SMI) is consistent with prior studies that validated CCR as a surrogate marker for muscle mass, particularly in older populations. In MASLD, characterized by systemic metabolic dysfunction, the co-occurrence of sarcopenia and fibrosis is clinically significant because muscle loss reflects overall disease burden and independently contributes to adverse outcomes, including mortality and hepatic decompensation [15]. Sarcopenia is frequently observed in patients with MASLD, with reported prevalence rates ranging from 20% to 40% depending on the study population and diagnostic criteria. The CCR serves as a marker of both skeletal muscle mass and strength and shows an indirect association with MASLD-related fibrosis, likely driven by progressive loss of muscle mass [16]. The link between muscle depletion and liver fibrosis involves shared pathways, including chronic inflammation, insulin resistance, mitochondrial dysfunction, and impaired anabolic signaling [15]. Moreover, the loss of protective, exercise-induced myokines—such as IL-6, irisin, and apelin, which have demonstrated protective effects against hepatic steatosis and fibrosis progression—along with the presence of hyperammonemia, may accelerate muscle atrophy and fibrogenesis in chronic liver disease [17]. In this context, CCR offers a more holistic reflection of disease severity than liver-focused markers alone, as it captures both hepatic and muscular status [17]. A key advantage of CCR is its accessibility: serum creatinine and cystatin C measurements are already widely available for estimating renal function in patients with metabolic disorders [18], allowing for CCR to be calculated without additional cost or specialized equipment. CCR and FIB-4 are both currently reimbursed by the national health insurance system, and ELF testing has recently been approved for the evaluation of chronic liver diseases. Within this clinical framework, CCR represents a cost-effective and readily accessible biomarker that could complement existing noninvasive tools for the detection of significant liver fibrosis (≥F2), thereby facilitating earlier risk stratification and management in routine clinical practice. Importantly, we demonstrated that CCR adds incremental diagnostic value to traditional fibrosis scores and, when combined with FIB-4 and ELF, enhances accuracy in identifying advanced fibrosis.

Nevertheless, several limitations must be acknowledged. The retrospective, single-center design, limited ethnic diversity, and the relatively small sample size may limit generalizability. Therefore, studies with larger and more diverse cohorts are required to validate these results. Although patients with advanced renal dysfunction were excluded, variations in creatinine and cystatin C levels can still arise from extrarenal factors, including inflammation, nutrition, and age-related muscle metabolism changes. Skeletal muscle mass was assessed using SMI, but functional measures of sarcopenia—such as handgrip strength or gait speed—were not evaluated, despite their inclusion in the Japan [19], European [20] and Asian [21] working group definitions. The length of the descriptions for diabetes mellitus (DM) treatment agents was not properly assessed. We also did not assess longitudinal changes in CCR, so its potential as a dynamic marker for tracking fibrosis progression or regression remains unknown. Future prospective, multicenter, and longitudinal studies are warranted to validate our findings, refine CCR cutoff thresholds for fibrosis detection, and explore its role in predicting key clinical outcomes, including hepatic decompensation, cardiovascular events, and mortality. Furthermore, integrating CCR into machine-learning models or composite diagnostic algorithms could enhance its clinical utility, enabling automated risk stratification within electronic health systems.

## 4. Materials and Methods

### 4.1. Study Design and Patients

This retrospective cohort study was conducted at the Nara Medical University Hospital between January 2018 and December 2023. We identified consecutive patients diagnosed with MASLD who underwent liver biopsy as part of the clinical evaluation for progressive liver disease. MASH is the histological, progressive form of metabolic dysfunction–associated steatotic liver disease (MASLD). The inclusion criteria were as follows: (1) histologically confirmed hepatic steatosis; and (2) at least one metabolic risk factor, including type 2 DM, hypertension, dyslipidemia, or obesity (body mass index, BMI ≥ 25 kg/m^2^ according to Japanese criteria). The exclusion criteria were as follows: (1) excessive alcohol intake (>30 g/day for men or >20 g/day for women) (n = 39); (2) coexisting chronic liver diseases, including viral hepatitis, autoimmune hepatitis, and drug-induced liver injury) (n = 8); (3) decompensated cirrhosis or hepatocellular carcinoma (n = 19); (4) significant renal impairment (eGFR < 30 mL/min/1.73 m^2^) (n = 19); and (5) systemic inflammatory conditions or active malignancies (to avoid confounding effects on serum creatinine and cystatin C levels) (n = 8) (Figure 10).

A total of 104 patients met all eligibility criteria and were included in the final analysis. The Institutional Review Board of Nara Medical University approved the study protocol on 14 November 2017 (No. 3521), which was conducted in accordance with the Declaration of Helsinki (2013) and renewed annually through 2018–2023. As this study involved a retrospective analysis of de-identified data, informed consent requirements were waived.

### 4.2. Clinical and Laboratory Data Collection

Demographic information—including age, sex, BMI, comorbidities, medication use, and alcohol consumption—was extracted from electronic medical records. Laboratory parameters were obtained from fasting blood samples collected within 2 weeks of liver biopsy and included liver function tests (i.e., AST, ALT, ALP, γ-GTP, bilirubin, and albumin), hematology (i.e., platelet count, white blood cell count, and hemoglobin), renal function (i.e., serum creatinine, cystatin C, and blood urea nitrogen), and metabolic profile (i.e., fasting plasma glucose, HbA1c, and lipid profile) [22].

Serum creatinine was measured using an enzymatic method calibrated to the isotope dilution mass spectrometry standard. Serum cystatin C levels were measured using particle-enhanced turbidimetric immunoassay. All assays were performed in a single centralized laboratory using standardized, quality-controlled procedures [23].

The CCR was calculated as follows:CCR = serum creatinine (mg/dL)/serum cystatin C (mg/L)Ref. [24]

The FIB-4 index was calculated as follows:FIB-4 = (Age × AST)/(Platelet count × √ALT) Ref. [25]

The D score was calculated as follows:ELF = 2.278 + 0.851 × ln(HA) + 0.751 × ln(PIIINP) + 0.394 × ln(TIMP-1)
where hyaluronic acid (HA), procollagen III N-terminal peptide (PIIINP), and tissue inhibitor of metalloproteinase-1 (TIMP-1) were measured using chemiluminescent immunoassay [26].

### 4.3. Liver Histology Assessment

Percutaneous liver biopsies were performed under ultrasound guidance using a 16-gauge needle. Adequate samples were defined as those ≥15 mm in length and containing at least 10 complete portal tracts. Specimens were fixed in formalin, embedded in paraffin, sectioned, and stained with hematoxylin–eosin and Masson’s trichrome [14].

Two experienced liver pathologists, blinded to all clinical and laboratory data, evaluated the samples. Fibrosis was staged according to the NASH Clinical Research Network system [23].

Disease activity was assessed using NAS, comprising steatosis (0–3), lobular inflammation (0–3), and ballooning degeneration (0–2) [24]. The total NAS score ranges from 0 to 8, with scores ≥ 5 indicative of steatohepatitis. In this study, advanced fibrosis was defined as stage ≥F2. Significant fibrosis was defined as stage ≥F2 according to the NASH Clinical Research Network system. Inter-observer variability in histological assessment was minimized through collaborative discussions between hepatologists and pathologists.

### 4.4. Assessment of Skeletal Muscle Mass

Skeletal muscle mass was assessed using CT imaging at the third lumbar vertebra (L3) level. Cross-sectional images were analyzed using ImageJ software 1.54p 17 February 2025 (upgrade). (NIH, Bethesda, MD, USA). The skeletal muscle area (cm^2^) was quantified by delineating the psoas, paraspinal, and abdominal wall muscles within predefined Hounsfield unit thresholds (−29 to +150 HU). SMI was calculated as follows:SMI = skeletal muscle area (cm^2^)/height^2^ (m^2^) Ref. [27]

Sarcopenia was defined according to Japanese sex-specific cutoff values, with thresholds <42 cm^2^/m^2^ for men and <38 cm^2^/m^2^ for women. All CT scans were performed within 3 months of liver biopsy [19].

### 4.5. Statistical Analyses

Statistical analyses were performed using IBM SPSS Statistics version 27.0 (Armonk, NY, USA). Continuous variables are expressed as mean ± standard deviation [28] or median with interquartile range (IQR), depending on the data distribution. The Shapiro–Wilk test was used to assess normality. Categorical variables are presented as frequencies and percentages [29].

Comparisons between groups, such as F < 2 vs. F ≥ 2, were made using Student’s *t*-test or the Mann–Whitney U test for continuous variables and the chi-square test or Fisher’s exact test for categorical variables. Pearson or Spearman correlation coefficients were used to assess the relationship between CCR and other continuous variables, including SMI and fibrosis stage [30].

Receiver operating characteristic [31] curves were generated to evaluate the discriminative ability of CCR, FIB-4, ELF, and their combinations for predicting advanced fibrosis (≥F2) [32,33]. The AUROC with 95% CIs was calculated, and optimal cutoff points were determined using the Youden index. Decision tree analysis was performed to classify patients with fibrosis stage >F2 using CCR, FIB-4, and ELF scores [34,35]. Sensitivity analyses were performed to assess the robustness of the CCR cutoff performance across different renal function strata and in subgroups with high versus low muscle mass. Missing data were addressed using listwise deletion, as missingness was <5% for all key variables.

## 5. Conclusions

CCR represents a promising, low-cost, and scalable biomarker that reflects both hepatic fibrosis and muscle mass in patients with MASLD who have preserved kidney function. Its combination with other noninvasive tools may improve fibrosis detection accuracy and support a more comprehensive, dual-focus assessment of metabolic liver disease. The lack of association with steatosis or inflammation underscores its specificity for chronic structural fibrotic changes and sarcopenia rather than acute hepatic injury.

## Figures and Tables

**Figure 1 ijms-26-09560-f001:**
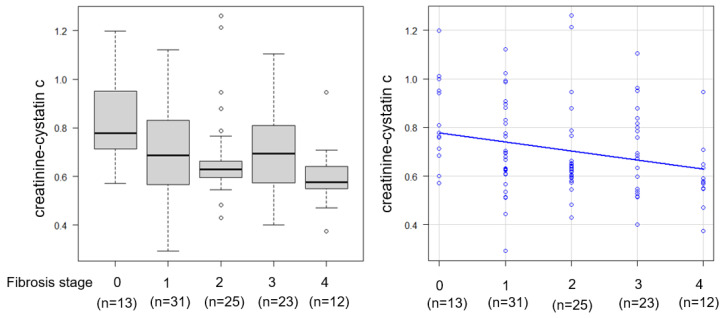
Association between the creatinine-to-cystatin C ratio (CCR) and liver fibrosis stage.

**Figure 2 ijms-26-09560-f002:**
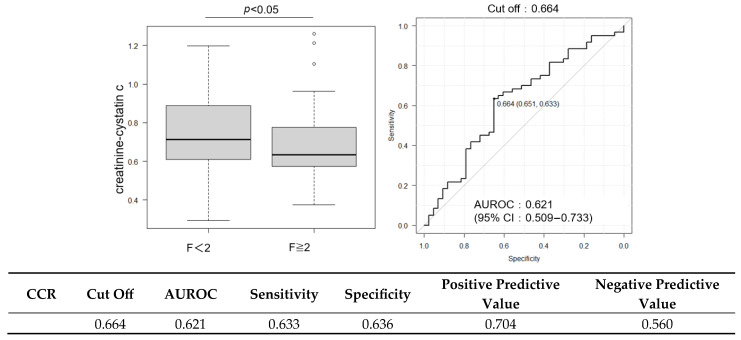
Receiver operating characteristic curve for CCR in predicting significant fibrosis (F ≥ 2). Cutoff value: 0.664; AUROC: 0.621 (95% CI: 0.509–0.733).

**Figure 3 ijms-26-09560-f003:**
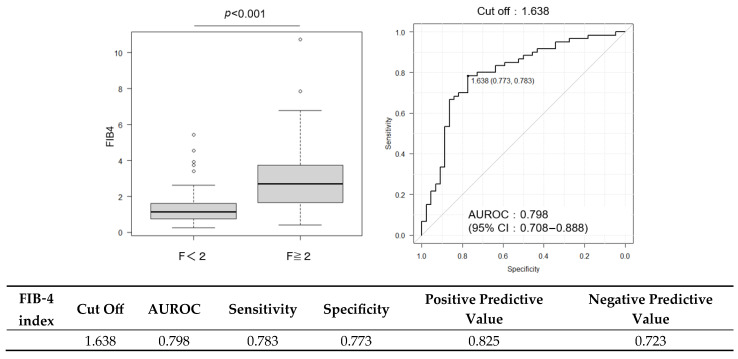
ROC curve for the FIB-4 index in predicting significant fibrosis (F ≥ 2). Cutoff value: 1.638; AUROC: 0.798 (95% CI: 0.708–0.888).

**Figure 4 ijms-26-09560-f004:**
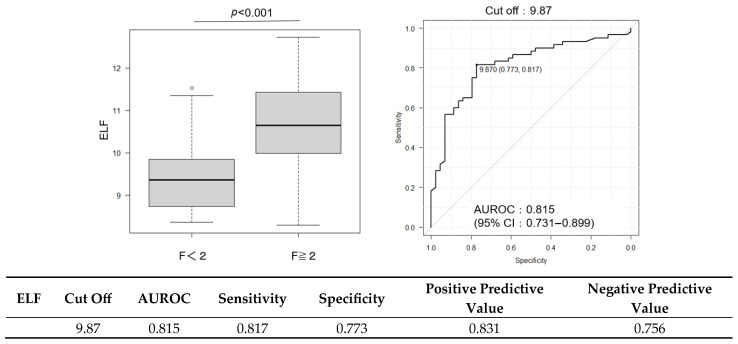
ROC curve for the ELF score in predicting significant fibrosis (F ≥ 2). Cutoff value: 9.87; AUROC: 0.815 (95% CI: 0.731–0.899).

**Figure 5 ijms-26-09560-f005:**
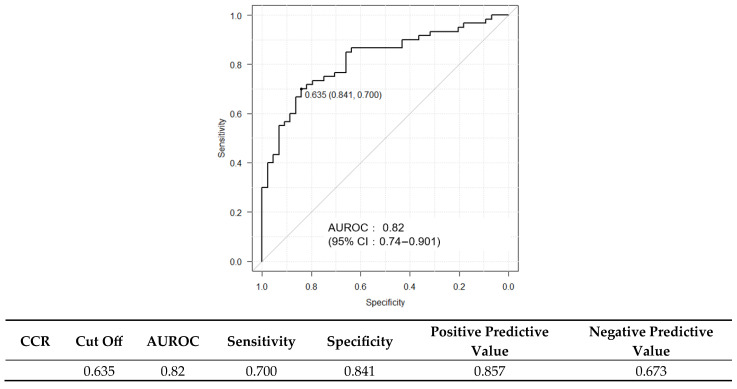
ROC curve for CCR combined with the FIB-4 index and ELF score in predicting significant fibrosis (F ≥ 2). Cutoff value: 1.638; AUROC: 0.798 (95% CI: 0.708–0.888).

**Figure 6 ijms-26-09560-f006:**
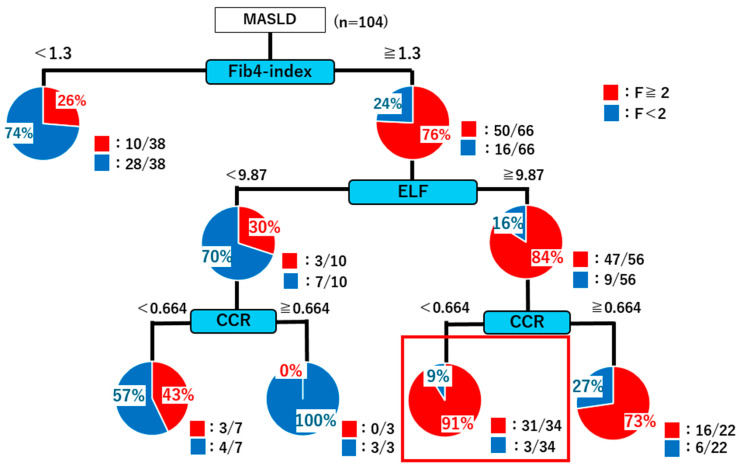
Decision tree analysis for identifying patients with significant fibrosis (F ≥ 2) based on the FIB-4 index, ELF score, and CCR.

**Figure 7 ijms-26-09560-f007:**
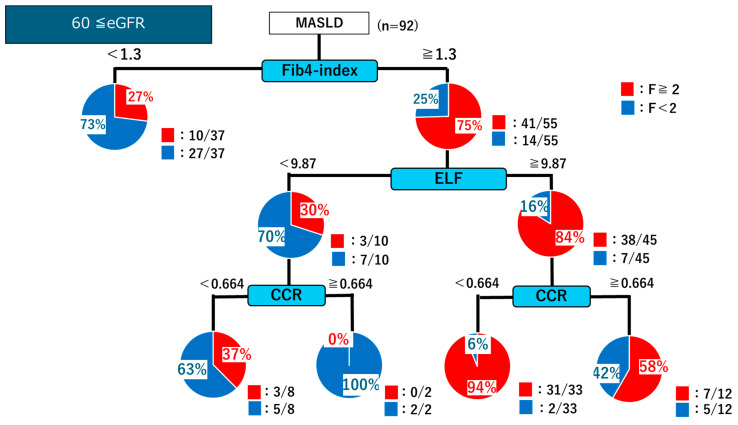
Decision tree analysis for identifying patients with significant fibrosis (F ≥ 2) based on the FIB-4 index, ELF score, and CCR in patients with eGFR ≥ 60.

**Figure 8 ijms-26-09560-f008:**
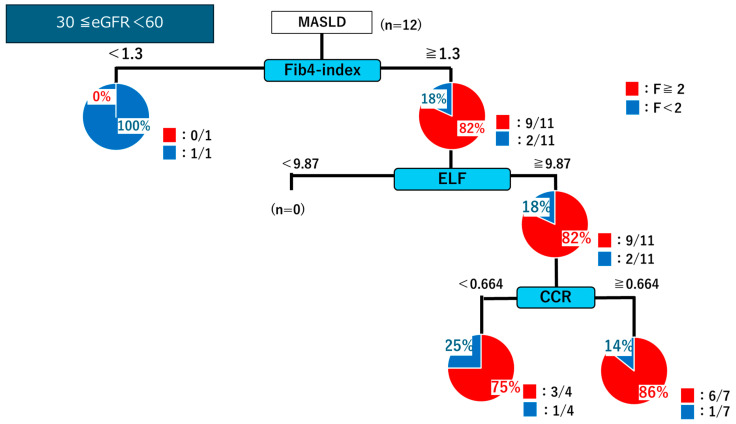
Decision tree analysis for identifying patients with significant fibrosis (F ≥ 2) based on the FIB-4 index, ELF score, and CCR in patients with eGFR of ≤30 to <60.

**Figure 9 ijms-26-09560-f009:**
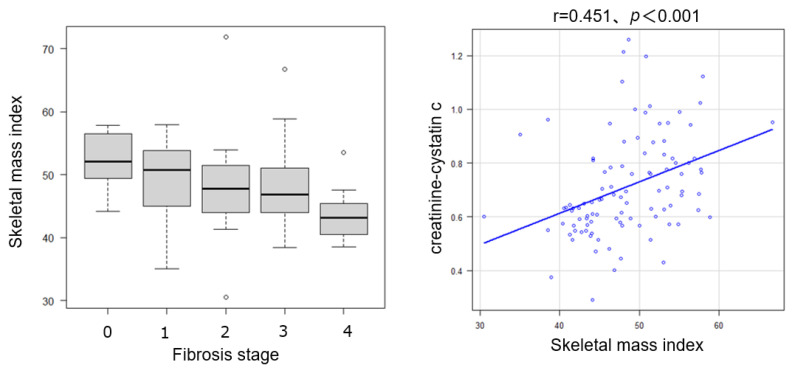
Correlation between CCR and the skeletal muscle index (SMI) (r = 0.451, *p* < 0.001).

**Figure 10 ijms-26-09560-f010:**
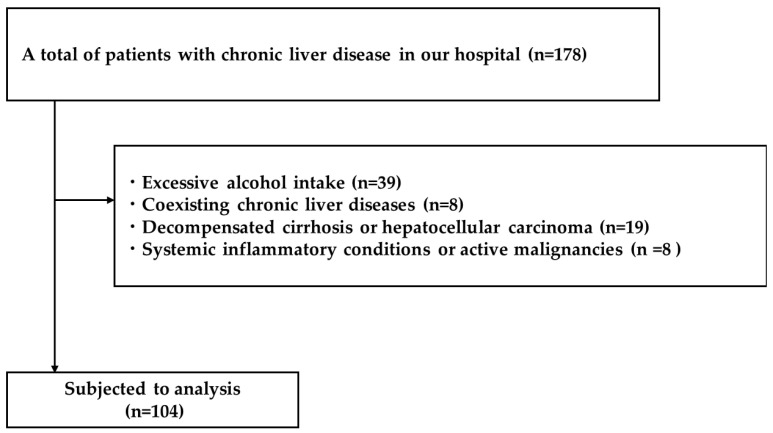
Flowchart of the current study.

**Table 1 ijms-26-09560-t001:** Clinical characteristics of the patients.

	All (n = 104)	F ≥ 2 (n = 60)	F < 2 (n = 44)	*p* Value
Age (years)	62.9 (50.0–67.8)	64.9 (55.9–68.1)	53.3 (34.9–65.0)	<0.001
Sex (M/F)	44/60	20/40	24/20	<0.05
BMI (kg/m^2^)	27.3 (24.6–31.2)	27.8 (24.4–31.9)	26.6 (25.1–30.0)	0.396
Plt (10^4^×/μL)	19 (15.8–24.1)	18.1 (14.7–21.1)	23.5 (18.2–27.7)	<0.001
AST (U/L)	48 (34–73)	54.5 (43.8–75.5)	38 (29.8–66.3)	<0.05
ALT (U/L)	64 (40–95)	61 (42.3–95)	65 (40–87)	0.508
γ-GT (U/L)	54 (37–91)	68.5 (45.8–95.5)	41.5 (27.8–65.3)	<0.05
T-bil (mg/dL)	0.8 (0.7–1.1)	0.8 (0.7–1.1)	0.8 (0.7–1.1)	0.573
ALB (g/dL)	4.3 (4.1–4.4)	4.2 (4–4.4)	4.4 (4.2–4.5)	<0.05
BUN (mg/dL)	14 (12–16)	14 (12–16)	13 (11–15)	0.0648
Cr (mg/dL)	0.69 (0.59–0.84)	0.68 (0.57–0.81)	0.71 (0.60–0.87)	0.394
Cysctatin C (mg/L)	1.03 (0.89–1.16)	1.04 (0.93–1.16)	1.01 (0.84–1.15)	0.185
HbA1c (%)	6.3 (5.7–7.0)	6.4 (5.8–7.2)	6.1 (5.6–6.9)	0.191
FIB-4 index	1.79 (1.08–3.19)	2.71 (1.66–3.70)	1.14 (0.75–2.04)	<0.001
Hypertension (no/yes)	48/56	24/36	24/20	0.145
Dyslipidemia (no/yes)	64/40	33/27	31/13	0.112
Diabetes (no/yes)	44/60	23/37	21/19	0.342
ELF score	10.06 (9.22–11.0)	10.65 (9.99–11.42)	9.35 (8.74–9.84)	<0.001
Cr/CysC (CCR)	0.658 (0.581–0.816)	0.634 (0.574–0.770)	0.736 (0.610–0.900)	<0.05
Fibrosis Stage	F0/F1/F2:F3/F4	F2/F3/F4:25/23/12	F0/F1:13/31	

Categorical data are presented as number, and continuous data as median value (interquartile range). BMI, body mass index; FIB-4 index, fibrosis-4 index; AST: Aspartate aminotransferase, ALT: Alanine aminotransferase, γ-GT: Gamma-glutamyl transferase, ALB: albumin, ELF score: Enhanced liver fibrosis score. T-bil stands for Total bilirubin. Cr/CysC refers to the creatinine-to-cystatin C ratio, Plt: platelet count.

## Data Availability

The datasets generated and/or analyzed during the current study are available from the corresponding author (T.N.) upon reasonable request.

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
