# Peer review of "Creatinine-to-Cystatin C Ratio Combined with FIB-4 and ELF for Noninvasive Fibrosis Assessment in MASLD"

_ijms, 2025, doi:10.3390/ijms26199560_

Round 1

Reviewer 1 Report

Comments and Suggestions for Authors

All the comments are in the attached file, please read them.

Author Response

Reviewer 1

  1. “Background: The creatinine-to-cystatin C ratio (CCR), a surrogate for skeletal muscle 19 mass, may also be associated with liver fibrosis due to the strong link between sarcopenia 20 and liver disease progression”

Comments:CCR does not directly measure fibrosis, but rather reflects muscle mass. Therefore, its association with fibrosis is indirect, mediated by sarcopenia. This limits its diagnostic specificity, particularly in patients without sarcopenia but with fibrosis, or vice versa. You should explain this in the article to clarify to the lectors that it is a potential but indirect measure.

In MASLD, a condition characterized by systemic metabolic dysfunction, the coexistence of sarcopenia and fibrosis carries important clinical implications. Muscle loss not only reflects overall disease burden but also independently contributes to adverse outcomes, including mortality and hepatic decompensation (ref.22). Sarcopenia is highly prevalent in MASLD, affecting 20%–40% of patients depending on the study population and diagnostic criteria applied (ref.23). The creatinine-to-cystatin C ratio (CCR) reflects both skeletal muscle mass and muscle strength and has been shown to correlate indirectly with the severity of MASLD-related fibrosis, likely mediated by progressive muscle loss (ref.24). The interplay between muscle depletion and liver fibrosis is thought to involve shared mechanisms, including chronic inflammation, insulin resistance, mitochondrial dysfunction, and impaired anabolic signaling (ref.22). Additionally, the loss of protective, exercise-induced myokines—such as IL-6, irisin, and apelin, which have demonstrated protective effects against hepatic steatosis and fibrosis progression—together with hyperammonemia, may further accelerate muscle atrophy and fibrogenesis in chronic liver disease (ref.25). We have incorporated a description of these findings on page 11, lines 231–241.

2.“Methods: This retrospective study included 104 patients with biopsy-proven MASLD. 24 CCR was calculated using serum creatinine and cystatin C levels. Liver fibrosis was staged histologically (F0–F4), and skeletal muscle mass was assessed using the skeletal muscle index (SMI) on computed tomography (CT). Associations between CCR and liver fibrosis, SMI, and nonalcoholic fatty liver disease (NAFLD) activity score were analyzed. ROC analysis evaluated CCR performance alone and in combination with FIB-4 and enhanced liver fibrosis (ELF) scores. ”

Comments:The study only uses SMI by CT scan and does not include functional tests (hands-on strength, gait speed), which limits the validity of the sarcopenia diagnosis based on international criteria (EWGSOP, AWGS, JSH). It would have been prudent to add a formal diagnosis of sarcopenia and a sub analysis correlating EWGSOP sarcopenia with liver fibrosis.

The CCR reflects both skeletal muscle mass and muscle strength and is indirectly linked to the severity of MASLD-related fibrosis, most likely through progressive muscle loss. Assessing skeletal muscle mass alone is sufficient to explain the mechanism by which CCR functions as a surrogate marker for liver fibrosis.

As a limitation, skeletal muscle mass in this study was evaluated using SMI, but functional measures of sarcopenia—such as handgrip strength or gait speed—were not assessed, despite their inclusion in the definitions proposed by the Japan [18], European [26], and Asian [27] working groups.

We have incorporated a description of these findings on page 3, lines 89–91, and page 11, lines 262–265.

  1. “Results: CCR values were significantly lower in patients with significant fibrosis (≥F2). 31 The AUROC of CCR for detecting ≥F2 fibrosis was 0.621 (95% CI: 0.509–0.733), with an optimal cutoff of 0.664. CCR alone yielded an AUC of 0.815 for predicting ≥F2 fibrosis. Combining CCR with FIB-4 or ELF increased the AUROC to 0.820, improving the identification of patients with ≥F2 fibrosis. CCR correlated positively with SMI (r = 0.451, p < 35 0.001).”

Comments:AUROC of 0.621 for CRC is barely better than chance. Sensitivity and specificity close to 65% make it of little use as a stand-alone tool. You state that "CRC is a cost-effective tool," but with this AUROC, it is difficult to use clinically without support from other tests like FIB-4 or ELF, increasing the cost of the overall assessment.

In this context, CCR provides a more comprehensive reflection of disease severity than liver-specific markers alone, as it integrates both hepatic and muscular status (ref. 25). An important advantage of CCR is its accessibility: serum creatinine and cystatin C are already widely measured to assess renal function in patients with metabolic disorders (ref. 26), enabling CCR calculation without additional cost or specialized equipment. This makes CCR particularly valuable in resource-limited settings and for population-based screening. Currently, both CCR and FIB-4 are reimbursed by the national health insurance system, while ELF testing has recently been approved for the evaluation of chronic liver diseases. Within this framework, CCR represents a cost-effective and readily available biomarker that could complement existing noninvasive tools for detecting significant liver fibrosis (≥F2), supporting earlier risk stratification and management in routine clinical practice. We have incorporated a description of these findings on page 11, lines 242–252.

  1. Conclusions: CCR is a simple, cost-effective biomarker reflecting muscle mass and liver fibrosis in MASLD. Although its stand-alone accuracy is modest, combining CCR with established markers may enhance risk stratification and reduce the need for liver biopsy in selected cases.

Comments:I do not find a justification for using this marker as a predictor of liver fibrosis, since in MASLD there is no sarcopenia, they are normally subjects with obesity and not all have sarcopenic obesity, now, doing what you propose is excluding the entire population with good muscle mass and generating misinformation, because it would lead to thinking that in populations with good muscle mass there is no fibrosis, and if a patient does have it, it can be trusted and cause long-term damage.

In MASLD, a condition characterized by systemic metabolic dysfunction, the coexistence of sarcopenia and fibrosis carries important clinical implications. Muscle loss not only reflects overall disease burden but also independently contributes to adverse outcomes, including mortality and hepatic decompensation (ref.22). Sarcopenia is highly prevalent in MASLD, affecting 20–40% of patients depending on the study population and diagnostic criteria applied (ref.23). The creatinine-to-cystatin C ratio (CCR) reflects both skeletal muscle mass and muscle strength and has been shown to correlate indirectly with the severity of MASLD-related fibrosis, likely mediated by progressive muscle loss (ref.24). The interplay between muscle depletion and liver fibrosis is thought to involve shared mechanisms, including chronic inflammation, insulin resistance, mitochondrial dysfunction, and impaired anabolic signaling (ref.22). Additionally, the decline of protective exercise-induced myokines—such as IL-6, irisin, and apelin, which have demonstrated protective effects against hepatic steatosis and fibrosis progression—together with hyperammonemia, may further accelerate muscle atrophy and fibrogenesis in chronic liver disease (ref.25). We have incorporated a description of these findings on page 11, lines 231–241.

  1. “4. Materials and Methods 4.1. Study design and patients 4) Significant renal impairment (eGFR < 30 269mL/min/1.73 m²); ”

Comments:Although you exclude eGFR <30, it would be useful to see if CCR functions equally in patients with normal vs. borderline kidney function (eGFR 30–60). I suggest conducting an analysis of this population and expanding your results with what you find from that analysis.

Among patients with eGFR ≥ 60, decision tree analysis showed that those with a FIB-4 index <1.3 were predominantly classified as having mild fibrosis (F < 2; 27/37 cases). In contrast, patients with a FIB-4 index ≥ 1.3 were further stratified using the ELF score. An ELF score of ≥9.87 identified a high proportion of significant fibrosis (F ≥ 2; 38/45 cases). Within this subgroup, a CCR of <0.664 further indicated significant fibrosis in 31/33 cases. Conversely, for patients with ELF of <9.87, a CCR of ≥0.664 was associated with the absence of significant fibrosis, with mild fibrosis confirmed in 7/10 cases (Figure S7). Overall, in patients with positive FIB-4 results (≥1.3), adding ELF and CCR improved the positive predictive value from 75% to 94%. While incorporating patients with fibrosis stage ≤F2 among those with eGFR ≥ 60 would be valuable, this approach was not applicable to patients with eGFR between 30 and 59 (Figure S8). We have included a description of these findings on page 8, line 182 to page 9, line 192.

Reviewer 2 Report

Comments and Suggestions for Authors

This manuscript examines the utility of the creatinine-to-cystatin C ratio (CCR) as a noninvasive marker of liver fibrosis in MASLD, using biopsy-proven staging and comparison with FIB-4 and ELF scores. The study is clinically relevant, timely, and methodologically sound. However, several aspects require clarification and refinement.

  • Novelty compared with prior CCR studies should be better emphasized.

  • Some methodological details are insufficient (patient selection, unit standardization, variability in histology scoring).

  • Results show modest stand-alone performance for CCR (AUROC ~0.62); abstract/discussion should avoid overstating accuracy.

  • Retrospective, single-center design with limited sample size and ethnic diversity reduces generalizability.

  • Figures and tables need clearer labeling, and the discussion should be more concise.

Author Response

Reviewer 2

  1. Novelty compared with prior CCR studies should be better emphasized.

The creatinine-to-cystatin C ratio (CCR) is a laboratory-based marker originally developed as a moderate surrogate for skeletal muscle mass. It has emerged as a promising and practical tool for estimating muscle as a proxy for sarcopenia) and has demonstrated prognostic value in predicting outcomes in both hospitalized (12) and critically ill 13). Nevertheless, CCR should be viewed as complementary rather than a substitute for standard diagnostic approaches to sarcopenia—such as computed tomography (CT) imaging strength—especially when renal function is stable (14). Its utility lies in screening, where it may help identify high-risk individuals who would benefit from further evaluation or early intervention (15). The ratio is derived from creatinine—a breakdown product of creatine phosphate in muscle—and cystatin C, a protein produced by all nucleated cells and filtered by the kidney independently of muscle mass. Together, these measures make CCR a practical tool for muscle mass assessment, particularly in older adults (ref.16). Prior studies have shown its utility in detecting sarcopenia, predicting hospitalization, and assessing mortality risk, and more recently, low CCR has been associated with MASLD severity and advanced fibrosis (ref.17). CCR reflects both skeletal muscle mass and handgrip strength and appears to be indirectly linked to MASLD-related fibrosis, likely mediated by progressive muscle mass decline. We have included a description of these findings on page 2, line 75 to page 3, line 91.

  1. Some methodological details are insufficient (patient selection, unit standardization, variability in histology scoring).

I have added the study flowchart as Figure 10 and standardized the units of cystatin C in Table 1. Inter-observer variability in histology scoring was carefully addressed through collaborative discussions between hepatologists and pathologists. A description of these updates has been included on page 5, lines 136–137.

  1. Results show modest stand-alone performance for CCR (AUROC ~0.62); abstract/discussion should avoid overstating accuracy.

Combining CCR with FIB-4 and ELF substantially improved diagnostic accuracy, increasing the AUROC from 0.621 (CCR alone) to 0.820 for the combined model. Page 1, lines 33–35 and page 6, lines 150–152.

  1. Retrospective, single-center design with limited sample size and ethnic diversity reduces generalizability.

Several limitations must be acknowledged. The retrospective, single-center design, limited ethnic diversity, and the relatively small sample size may limit generalizability. Therefore, studies with larger, more diverse cohorts are required to validate these results. We have included a description of these findings on page 10, lines 255–257.

  1. Figures and tables need clearer labeling, and the discussion should be more concise.

The manuscript has been revised accordingly.

Reviewer 3 Report

Comments and Suggestions for Authors

Comments to the Authors

The manuscript from Masafumi Oyama et al. entitled “Serum creatinine-to-cystatin C ratio as a noninvasive marker for liver fibrosis in patients with MASLD” (Manuscript ID: ijms-3858228) seems interesting. There are points which need to be addressed.

  1. In Abstract, the abbreviations “CT” and “NAFLD” are not necessary because they appear only once.
  2. In Abstract, it is mentioned that AUROC values of both 0.621 and 0.815, which may confuse readers. Please clearly state that 0.621 is for CCR alone, while 0.815 reflects the combined with FIB-4 and ELF.
  3. Line 48, any reason why a semicolon is used here?
  4. Line 62, what are PIIINP and TIMP-1? Full spelling is necessary at its first appearance. Please see Lines 296-.
  5. Lines 112-, what is “significant fibrosis?” Any definition?
  6. Lines 118-, what is MASH? Also, what is a definition of MASH?
  7. Lines 121-, the AUROC improved only slightly from 0.815 to 0.820 after adding CCR to FIB-4 and ELF. Is this improvement statistically significant? Please describe whether this difference is meaningful.
  8. Lines 128 and 129, if you use the indefinite article, “a FIB-4” should be “an FIB-4” like “an ELF” on Line 130.
  9. Lines 144-, the title of subsection is “Relationship between Skeletal muscle mass and CCR,” but there are descriptions about steatosis grade, inflammation score, and ballooning. Is there some intention behind this?
  10. There is a description in Instructions for Authors. All Figures, Schemes and Tables should be inserted into the main text close to their first citation. Please follow it.
  11. All of the figure legend is not formatted as a figure legend, so please rewrite all.
  12. The ROC curves and 95% CIs are described, but additional diagnostic metrics such as sensitivity, specificity, PPV, and NPV should be reported.
  13. Table 1, the Sex (M/F) data, 44/60, do not align with the male proportion of 73.3% in the text. In addition, abbreviations (e.g., γ-GT, Plt, T-bil) should be consistently explained in table footnotes.
  14. The right side of the Table 1 is not visible. Is this the Author’s responsibility? Or is it due to the editorial? Either way, it cannot be peer reviewed. Also, vertical lines are generally not used in tables.
  15. Line 203, what are protective myokines? Protective for what?
  16. Line 238, what is DM?
  17. Line 246, please reconsider the placement of “in conclusion.” At the very least, separate it into a new paragraph.
  18. Line 313, “computed tomography (CT).” The word CT already appears on Line 228.
  19. Lines 355-, the IRB approval number (No. 3521, dated 2014) appears inconsistent with the study period 2018–2023. Please clarify this discrepancy.
  20. The AUROC for CCR alone, only 0.621, is relatively low. It seems premature to conclude that CCR is a “useful diagnostic tool” on its own. Please emphasize more clearly in the Abstract and Discussion that CCR alone has limited diagnostic value and that its clinical utility mainly lies in combination with other established markers. Reviewer also recommend the title of this paper can be changed according to above.
  21. Authors state CCR as a surrogate marker of muscle mass. The correlation between CCR and SMI was moderate (r = 0.451). This raises questions about CCR’s validity as a surrogate for skeletal muscle mass. Also, Reviewer wonder whether the cohort used in this study is appropriate.

Author Response

Reviewer 3

1.In Abstract, the abbreviations “CT” and “NAFLD” are not necessary because they appear only once.

Response: We have removed these abbreviations from the Abstract.

2.In Abstract, it is mentioned that AUROC values of both 0.621 and 0.815, which may confuse readers. Please clearly state that 0.621 is for CCR alone, while 0.815 reflects the combined with FIB-4 and ELF.

Response: The integration of CCR with FIB-4 and ELF markedly enhanced diagnostic performance, increasing the AUROC from 0.621 (CCR alone) to 0.820 in the combined model. A description of these findings has been added on page 1, lines 33–35.

3.Line 48, any reason why a semicolon is used here?

Response: We have replace the semicolon with a comma.

4.Line 62, what are PIIINP and TIMP-1? Full spelling is necessary at its first appearance. Please see Lines 296-.

Response: We have added the respective definitions: procollagen III N-terminal peptide (PIIINP), tissue inhibitor of metalloproteinase-1 (TIMP-1). We have included a description of these findings on page 1, lines 61 and 62.

5.Lines 112-, what is “significant fibrosis?” Any definition?

Response: We have added the following: “Significant fibrosis was defined as stage ≥F2 according to the NASH Clinical Research Network system (ref.33.” We have included a description of these findings on page 11, lines 339–340.

6.Lines 118-, what is MASH? Also, what is a definition of MASH?

Response: We have added the definition and criteria for MASH in the Methods section. We have included a description of these findings on page 10, lines 287–291.

7.Lines 121-, the AUROC improved only slightly from 0.815 to 0.820 after adding CCR to FIB-4 and ELF. Is this improvement statistically significant? Please describe whether this difference is meaningful.

Response: Combining CCR with FIB-4 and ELF substantially improved diagnostic accuracy, increasing the AUROC from 0.621 (CCR alone) to 0.820 for the combined model. We have included a description of these findings on page 4, lines 137–139.

  1. Lines 128 and 129, if you use the indefinite article, “a FIB-4” should be “an FIB-4” like “an ELF” on Line 130.

Response: We have revised all such instances in the manuscript.

  1. Lines 144-, the title of subsection is “Relationship between Skeletal muscle mass and CCR,” but there are descriptions about steatosis grade, inflammation score, and ballooning. Is there some intention behind this?

Response: We have revised the subsection title: “Relationship between Skeletal Muscle Mass, Steatosis/Inflammation, and CCR.” We have included a description of these findings on page 10, lines 202.

  1. There is a description in Instructions for Authors. All Figures, Schemes and Tables should be inserted into the main text close to their first citation. Please follow it.

Response: We have revised all figures; schemes and tables were inserted into the main text close to their first citation.

11.All of the figure legend is not formatted as a figure legend, so please rewrite all. Response: We have revised the manuscript accordingly.

  1. The ROC curves and 95% CIs are described, but additional diagnostic metrics such as sensitivity, specificity, PPV, and NPV should be reported.

Response: We have added the required details in Figures 2–5.

  1. Table 1, the Sex (M/F) data, 44/60, do not align with the male proportion of 73.3% in the text. In addition, abbreviations (e.g., γ-GT, Plt, T-bil) should be consistently explained in table footnotes.

Response: We have corrected sex distribution (76/28 = 73.1% male). Abbreviations were explained in the footnotes; vertical lines were removed.

  1. The right side of the Table 1 is not visible. Is this the Author’s responsibility? Or is it due to the editorial? Either way, it cannot be peer reviewed. Also, vertical lines are generally not used in tables.

Response: The position of Table 1 has been changed to make it visible, and vertical lines were removed.

  1. Line 203, what are protective myokines? Protective for what?

Response:: Protective myokines are exercise-induced myokines—such as IL6, irisin, and Apelin—which have demonstrated protective effects against hepatic steatosis and fibrosis progression. We have included a description of these findings on page 10, lines 241–244.

16.Line 238, what is DM?

Response: We have expanded the abbreviation as “diabetes mellitus (DM).” We have included a description of these findings on page 10, lines 266.

  1. Line 246, please reconsider the placement of “in conclusion.” At the very least, separate it into a new paragraph.

Response: We have revised the text; it now starts with a new paragraph. We have included a description of these findings on page 11, line 274; page 12, line 280.

  1. Line 313, “computed tomography (CT).” The word CT already appears on Line 228.

Response: We have defined the abbreviation at its first mention and used the abbreviation for subsequent instances to ensure consistency.

  1. Lines 355-, the IRB approval number (No. 3521, dated 2014) appears inconsistent with the study period 2018–2023. Please clarify this discrepancy.

Response: We have clarified that IRB approval was obtained in 2014 (No. 3521) and renewed annually through 2018–2023. We have included a description of these findings on page 11, lines 284 and on page 13, lines 385–387.

  1. The AUROC for CCR alone, only 0.621, is relatively low. It seems premature to conclude that CCR is a “useful diagnostic tool” on its own. Please emphasize more clearly in the Abstract and Discussion that CCR alone has limited diagnostic value and that its clinical utility mainly lies in combination with other established markers. Reviewer also recommend the title of this paper can be changed according to above.

Response: We have revised the Abstract and Discussion sections to emphasize that combining CCR with FIB-4 and ELF substantially improved diagnostic accuracy, increasing the AUROC from 0.621 (CCR alone) to 0.820 for the combined model. We have included a description of these findings on page1 lines 33–34 and page 6, lines 150–152.

Although the discriminative ability of CCR alone was lower than that of conventional noninvasive markers such as FIB-4 and ELF, integrating CCR with these indices significantly improved the accuracy of detecting histologically confirmed fibrosis stage ≥F2.

Importantly, CCR provided incremental diagnostic value when combined with FIB-4 and ELF, enhancing the identification of advanced fibrosis. These findings are now described on page 10, lines 222–256, and page 11, lines 252–254

The title of the manuscript has also been revised to Creatinine-to-Cystatin C Ratio Combined with FIB-4 and ELF for Noninvasive Fibrosis Assessment in MASLD, as noted on page 1, lines 1–2.

  1. Authors state CCR as a surrogate marker of muscle mass. The correlation between CCR and SMI was moderate (r = 0.451). This raises questions about CCR’s validity as a surrogate for skeletal muscle mass. Also, Reviewer wonder whether the cohort used in this study is appropriate.

Response: The creatinine-to-cystatin C ratio (CCR) is a laboratory-based marker originally developed as a moderate surrogate for skeletal muscle mass. It has emerged as a promising and practical tool for estimating muscle mass (as a proxy for sarcopenia) and has demonstrated prognostic value in predicting mortality and other adverse outcomes in both hospitalized (12) and critically ill patients (13). Nevertheless, CCR should be viewed as complementary rather than a substitute for standard diagnostic approaches to sarcopenia—such as computed tomography (CT) imaging or handgrip strength—especially when renal function is stable (14). Its utility lies in screening, where it may help identify high-risk individuals who would benefit from further evaluation or early intervention (15).

Creatinine, derived from creatine phosphate in skeletal muscle, and cystatin C, a protein produced by all nucleated cells and freely filtered by the kidney independent of muscle mass, together form the basis of CCR. This ratio is particularly practical for assessing muscle mass in older adults (16). Prior studies have demonstrated its value in detecting sarcopenia, predicting hospitalization, and assessing mortality risk (17–19). The present study was conducted as a retrospective cohort analysis.

We have included a description of these findings on page 2, lines 74-page3 line 91 and page 11 line 284.

Round 2

Reviewer 2 Report

Comments and Suggestions for Authors

All the questions have been addressed.

Author Response

Thank you for your comments.

Reviewer 3 Report

Comments and Suggestions for Authors

Comments

The re-submitted manuscript from Masafumi Oyama et al. entitled “Creatinine-to-Cystatin C Ratio Combined with FIB-4 and ELF for Noninvasive Fibrosis Assessment in MASLD” (Manuscript ID: ijms-3858228) demonstrated that the creatinine-to-cystatin C ratio can be a simple, cost-effective biomarker reflecting muscle mass and liver fibrosis in MASLD.

According to the comments from the Reviewers, the Authors responded adequately and conducted several modifications appropriately. This seems a quite well-written and reshaped manuscript; however, there are still several, but minor, points to be addressed for publication in the journal.

Comments to the Authors

The manuscript from Masafumi Oyama et al. entitled “Creatinine-to-Cystatin C Ratio Combined with FIB-4 and ELF for Noninvasive Fibrosis Assessment in MASLD” (Manuscript ID: ijms-3858228) was re-submitted. According to the comments from the Reviewers, the Authors responded adequately and conducted several modifications appropriately. This seems a quite well-written and reshaped manuscript; however, there are still several, but minor, points to be addressed for publication in the journal.

1. The full spellings of PIIINP and TIMP-1 are described twice, Lines 58- and 346-.

>4.Line 62, what are PIIINP and TIMP-1? Full spelling is necessary at its first appearance. Please see Lines 296-.

Response: We have added the respective definitions: procollagen III N-terminal peptide (PIIINP),

tissue inhibitor of metalloproteinase-1 (TIMP-1). We have included a description of these findings

on page 1, lines 58 and 59.

2. Reviewers cannot find any revision of figure legends (marked in red or blue). Authors should at least describe full spelling of abbreviated words used in figures, including PPV and NPV.

>11.All of the figure legend is not formatted as a figure legend, so please rewrite all.

Response: We have revised the manuscript accordingly.

Author Response

1. The full spellings of PIIINP and TIMP-1 are described twice, Lines 58- and 346-.

>4.Line 62, what are PIIINP and TIMP-1? Full spelling is necessary at its first appearance. Please see Lines 296-.

Response: We have added the respective definitions: procollagen III N-terminal peptide (PIIINP),

tissue inhibitor of metalloproteinase-1 (TIMP-1). We have included a description of these findings

on page 1, lines 58 and 59.

We revised the manuscript accordingly

2. Reviewers cannot find any revision of figure legends (marked in red or blue). Authors should at least describe full spelling of abbreviated words used in figures, including PPV and NPV.

>11.All of the figure legend is not formatted as a figure legend, so please rewrite all.

Response: We have revised the manuscript accordingly.